# Bayesian Multi-type Mean Field Multi-agent Imitation Learning

**Fan Yang**
Computer Science and Engineering
University at Buffalo
fyang24@buffalo.edu

**Alina Vereshchaka**
Computer Science and Engineering
University at Buffalo
avereshc@buffalo.edu

**Changyou Chen**
Computer Science and Engineering
University at Buffalo
changyou@buffalo.edu

**Wen Dong**
Computer Science and Engineering
University at Buffalo
wendong@buffalo.edu

## Abstract

Multi-agent Imitation learning (MAIL) refers to the problem that agents learn to perform a task interactively in a multi-agent system through observing and mimicking expert demonstrations, without any knowledge of a reward function from the environment. MAIL has received a lot of attention due to promising results achieved on synthesized tasks, with the potential to be applied to complex real-world multi-agent tasks. Key challenges for MAIL include sample efficiency and scalability. In this paper, we proposed Bayesian multi-type mean field multi-agent imitation learning (BM3IL). Our method improves sample efficiency through establishing a Bayesian formulation for MAIL, and enhances scalability through introducing a new multi-type mean field approximation. We demonstrate the performance of our algorithm through benchmarking with three state-of-the-art multi-agent imitation learning algorithms on several tasks, including solving a multi-agent traffic optimization problem in a real-world transportation network. Experimental results indicate that our algorithm significantly outperforms all other algorithms in all scenarios.

## 1 Introduction

Multi-agent imitation learning tries to infer a hidden reward function from expert demonstrations and optimizes a policy with the learned reward function for each agent in a multi-agent system. MAIL has received a lot of research attention and shown promising results over a variety of tasks, including the particle environments [15] and cooperative robotic control tasks based on OpenAI baselines [2], social group communication [17], driving simulation [1], sports [10, 11], and *etc*. [18] formulates the multi-agent generative adversarial imitation learning (MA-GAIL) by extending the single-agent generative adversarial imitation learning [4] to the multi-agent case. [7] incorporates MA-GAIL with an attention-actor-critic [6] to develop a multi-agent discriminator-actor-attention-critic (MA-DAAC) algorithm.

MAIL has the potential to solve real-world complex problems, such as optimizing driving routes in a city-scale transportation network [19], or optimizing the medical resource allocation to mitigate the spread of disease [9]. However, applying MAIL to real-world problems is not easy. MAIL is often formulated as solving a Markov game with unknown reward functions [18, 7]. To find the optimal policy in a Markov game, each agent needs to take into consideration the policies of other agents [21]. The more agents in the environment, the more variance and uncertainty each agent has in the policy

search. Moreover, interacting with the environment to collect samples is an expensive operation [20]. As such, the two fundamental challenges are sample efficiency and scalability.

To improve the sample efficiency, we introduce a Bayesian approach for MAIL which learns a more stable reward function to more efficiently guide the policy search, and which enables an algorithm to converge faster. To improve the scalability, we introduce a new multi-type mean field approximation to effectively gather the information from other agents, which approximates the interactions within the population of agents with those between a single agent and the average effect of the overall population. To this end, we develop a new imitation learning algorithm, the Bayesian multi-type mean field multi-agent imitation learning (BM3IL).

The contributions of the paper are summarized as follows. (1) We introduce a Bayesian formulation of MAIL, which could improve the sample efficiency and convergence speed. (2) We introduce a new multi-agent mean field approximation, which is more flexible and can achieve a better approximation comparing to existing multi-type mean field approximation. (3) We apply the multi-agent mean field approximation to a Bayesian formulation of multi-agent imitation learning and derive BM3IL, which is both sample efficient and scalable to complex environments. (4) We demonstrate empirical performance through benchmarking with existing algorithms in several scenarios, including real-world city-scale transportation networks.

## 2 Background

### 2.1 Markov games and Nash equilibrium

A Markov game with $N$ agents is formalized by the tuple G$(\mathcal{S}, \mathcal{A}^1, \ldots, \mathcal{A}^N, r^1, \ldots, r^N, p, \gamma)$, where $\mathcal{S}$ is a state space, $\mathcal{A}^i$ is the action space of agent $i \in \{1, \ldots, N\}$. The reward function for agent $i$ is defined as $r^i : \mathcal{S} \times \mathcal{A}^1 \times \ldots \times \mathcal{A}^N \to \mathbb{R}$. $p$ is the transition probability $\mathcal{S} \times \mathcal{A}^1 \times \ldots \times \mathcal{A}^N \to \Omega(\mathcal{S})$, with $\Omega(\mathcal{S})$ being the collection of probability distributions over the state space $S$. $\gamma \in [0, 1)$ is a discount factor. At time step $t$, all agents take actions simultaneously, each receiving an immediate reward $r_t^i$, or equivalently, the cost $c_t^i = -r_t^i$. The initial states are determined by a distribution $\eta : S \to [0, 1]$. The joint policy is defined as $\boldsymbol{\pi}(\boldsymbol{a}|s) = \prod_{i=1}^N \pi_i(a^i|s)$, where we use bold symbols to denote the concatenation of all variables for all agents. For agent $i$, the corresponding policy is defined as $\pi^i : S \to \Omega(\mathcal{A}^i)$, where $\Omega(\mathcal{A}^i)$ is the collection of probability distributions over agent $i$'s action space $\mathcal{A}^i$. We use $\pi^{-i} = [\pi^1, ..., \pi^{i-1}, \pi^{i+1}, ..., \pi^N]$ to represent the joint policy except $\pi^i$. Provided an initial state $s$, the value function of agent $i$ under the joint policy $\boldsymbol{\pi}$ is defined as $V_{\boldsymbol{\pi}}^i(s) = \sum_{t=0}^\infty \gamma^t \mathbb{E}_{\boldsymbol{\pi},p}[r_t^i|s_0 = s, \boldsymbol{\pi}]$. The Q function $Q_{\boldsymbol{\pi}}^i : \mathcal{S} \times \mathcal{A}^1 \times \cdots \times \mathcal{A}^n \to \mathbb{R}$ of agent $i$ under the joint policy $\pi$ can be formulated as $Q_{\boldsymbol{\pi}}^i(s, \boldsymbol{a}) = r^i(s, \boldsymbol{a}) + \gamma \mathbb{E}_{s' \sim p}[V_{\boldsymbol{\pi}}^i(s')]$, where $s'$ is the state at the next time step. The occupancy measure $\rho_{\boldsymbol{\pi}}(s, \boldsymbol{a}) = \sum_t \gamma^t p(s_t = s, \boldsymbol{a_t} = \boldsymbol{a})$ defines the state action visitation distribution using the joint policy $\boldsymbol{\pi}$.

Nash equilibrium [5] describes the situation that all agents can not improve their value through changing its own policy. It is represented by a joint policy $\boldsymbol{\pi}^* = [\pi^{1,*}, \pi^{2,*}, ..., \pi^{N,*}]$ such that $V_{\boldsymbol{\pi}}^i(s) \geq V_{\pi^i, \pi^{-i,*}}(s), \forall i, \forall \pi^i, \forall s$. Let $\boldsymbol{V}^*(s) = [V_{\boldsymbol{\pi}^*}^1(s), V_{\boldsymbol{\pi}^*}^2(s), ..., V_{\boldsymbol{\pi}^*}^N(s)]$ be the Nash value function associated with the Nash policy $\boldsymbol{\pi}^*$ , Nash Q-learning [5] defines a Q iteration algorithm $\mathscr{H}^{\text{Nash}} \boldsymbol{Q}(s, \boldsymbol{a}) = \mathbb{E}_{s'}[\boldsymbol{r}(s, \boldsymbol{a}) + \gamma \boldsymbol{V}^*(s')]$, where $\mathscr{H}^{\text{Nash}}$ is the Nash operator, $\boldsymbol{Q} = [Q^1, Q^2, ..., Q^N]$ is the Q function for all agents, and $\boldsymbol{r}(s, \boldsymbol{a}) = [r^1(s, \boldsymbol{a}), r^2(s, \boldsymbol{a}), ..., r^N(s, \boldsymbol{a})]$ is the reward function for all agents. Given certain assumptions [5] the Q function will eventually converge to a Nash equilibrium, which is referred to as the Nash Q value $\boldsymbol{Q}^* = [Q^{1,*}, Q^{2,*}, ..., Q^{N,*}]$.

### 2.2 Multi-agent generative adversarial imitation learning

In a Markov game G$(\mathcal{S}, \mathcal{A}^1, \ldots, \mathcal{A}^N, r^1, \ldots, r^N, p, \gamma)$, let $\boldsymbol{\theta} = \{\theta^i\}_{i=1}^N$ be the parameters associated with a policy, and $\boldsymbol{\phi} = \{\phi^i\}_{i=1}^N$ the parameters associated with a reward function. The target function of MA-GAIL is $\arg \min_{\boldsymbol{\theta}} \max_{\boldsymbol{\phi}} \mathbb{E}_{s, \boldsymbol{a} \sim \rho_{\pi_\theta}} \left[ \sum_{i=1}^N \log D(s, a^i; \phi^i) \right] + \mathbb{E}_{s, \boldsymbol{a} \sim \rho_{\pi_E}} \left[ \sum_{i=1}^N \log(1 - D(s, a^i; \phi^i)) \right]$, which can be solved by iterating between optimizing the

reward parameters

$$\arg\max_{\phi}\mathbb{E}_{s,\boldsymbol{a}\sim\rho_{\pi_{\theta}}}\left[\sum_{i=1}^{N}\log D(s,a^i;\phi^i)\right] + \mathbb{E}_{s,\boldsymbol{a}\sim\rho_{\pi_E}}\left[\sum_{i=1}^{N}\log(1-D(s,a^i;\phi^i))\right] \quad (1)$$

and optimizing the policy parameters, where $D$ is a discriminative classifier

$$\arg\min_{\boldsymbol{\theta}}\mathbb{E}_{s,\boldsymbol{a}\sim\rho_{\pi_{\theta}}}\left[\sum_{i=1}^{N}\log D(s,a^i;\phi^i)\right] \quad (2)$$

## 3 Methodology

We first reformulate multi-agent imitation learning in a Bayesian learning framework; then develop a new hierarchical multi-type mean field approximation. Finally, the algorithm we proposed, BM3IL, is introduced in details. The derivations and proofs are postponed to the Appendix.

### 3.1 Bayesian multi-agent imitation learning

#### 3.1.1 A probabilistic view of multi-agent imitation learning

In a Markov game $G(\mathcal{S}, \mathcal{A}^1, \ldots, \mathcal{A}^N, r^1, \ldots, r^N, p, \gamma)$, we introduce binary observation variables for the expert and agent, $\boldsymbol{o_E} = \{\{o_{E,t}^i\}_{i=1}^N\}_{t=1}^T$ and $\boldsymbol{o_A} = \{\{o_{A,t}^i\}_{i=1}^N\}_{t=1}^T$, for each agent $i$ and time step $t$. The subscript "$E$" represents an expert, "$A$" represent an agent. This convention will be used in the following, and sometimes the subscripts are dropped when there is no confusion. The probability of $o_t^i$ being equal to 1 is defined to be proportional to the exponential of a negative reward function: $p(o_t^i = 1 \mid s_t, a_t^i) \propto \exp\left(c(s_t, a_t^i)\right) = \exp\left(-r(s_t, a_t^i)\right)$. The graphical representation is shown in Figure 1. Let the policy be parameterized by $\boldsymbol{\theta} = \{\theta^i\}_{i=1}^N$, the reward function parameterized by $\boldsymbol{\phi} = \{\phi^i\}_{i=1}^N$, and denote the trajectory of state action pairs as $\boldsymbol{\tau} = \{\{s_t, a_t^i\}_{i=1}^N\}_{t=1}^T$. Regarding the parameters $\boldsymbol{\phi}, \boldsymbol{\theta}$ as random variables, the probability measure of agent and expert trajectories with observations can then be represented as

$$p(\boldsymbol{\tau_E}, \boldsymbol{\tau_A}, \boldsymbol{o_E}, \boldsymbol{o_A}, \boldsymbol{\theta}, \boldsymbol{\phi}) = p(\boldsymbol{\theta})p(\boldsymbol{\phi})p(\boldsymbol{\tau_E})p(\boldsymbol{\tau_A};\boldsymbol{\theta})\prod_{t=1}^{T}\prod_{i=1}^{N}p(o_{E,t}^i \mid s_{E,t}, a_{E,t}^i; \phi^i)p(o_{A,t}^i \mid s_{A,t}, a_{A,t}^i; \phi^i)$$

where $p(\boldsymbol{\tau_A};\boldsymbol{\theta}) = p(s_{A,0})\prod_{t=1}^{T}\left(\prod_{i=1}^{N}p(a_{A,t}^i \mid s_{A,t};\theta^i)\right)p(s_{A,t+1} \mid s_{A,t}, a_{A,t}^1, ..., a_{A,t}^N; \gamma)$. Here $p(s_{A,t+1} \mid s_{A,t}, a_{A,t}^1, ..., a_{A,t}^N; \gamma)$ means that with probability $\gamma$, the transition probability $p(s_{A,t+1} \mid s_{A,t}, a_{A,t}^1, ..., a_{A,t}^N)$ will happen as the dynamics defined in a Markov game. With probability $1 - \gamma$, regardless of actions, it will transit into an absorbing state with reward zero (no observation in absorbing state). For each agent $i$, we estimate the observation probability $p(o_t^i \mid s_t, a_t^i; \phi^i)$ with a discriminator $D(s_t, a_t^i; \phi^i)$ such that $p(o_t^i = 1 \mid s_t, a_t^i; \phi^i) = D(s_t, a_t^i; \phi^i)$ and $p(o_t^i = 0 \mid s_t, a_t^i; \phi^i) = 1 - D(s_t, a_t^i; \phi^i)$.

Based on this, Eq. 1 could be viewed as finding a single point estimation of the parameter $\phi$ that maximizes a surrogate objective of the log posterior $\log p(\boldsymbol{\phi} \mid \boldsymbol{o_E} = 0, \boldsymbol{o_A} = 1, \boldsymbol{\theta})$, conditioned on seeing observation $o_{E,t}^i = 0$ for expert samples and $o_{A,t}^i = 1$ for agents samples. The interpretation is to maximize the reward of expert samples and meanwhile to minimize the reward of agents samples.

$$\log p(\boldsymbol{\phi} \mid \boldsymbol{o_E} = 0, \boldsymbol{o_A} = 1, \boldsymbol{\theta})$$
$$\propto \log\mathbb{E}_{\tau_E, \tau_A}\prod_{t=1}^{T}\prod_{i=1}^{N}p(o_{E,t}^i = 0 \mid s_{E,t}, a_{E,t}^i; \phi^i)p(o_{A,t}^i = 1 \mid s_{A,t}, a_{A,t}^i; \phi^i)$$
$$\geq \mathbb{E}_{\tau_E, \tau_A}\log\prod_{t=1}^{T}\prod_{i=1}^{N}p(o_{E,t}^i = 0 \mid s_{E,t}, a_{E,t}^i; \phi^i)p(o_{A,t}^i = 1 \mid s_{A,t}, a_{A,t}^i; \phi^i) \quad (3)$$
$$= \mathbb{E}_{s_A, \boldsymbol{a}_A\sim\rho_{\pi_{\theta}}}\left[\sum_{i=1}^{N}\log D(s_A, a_A^i; \phi^i)\right] + \mathbb{E}_{s_E, \boldsymbol{a}_E\sim\rho_{\pi_E}}\left[\sum_{i=1}^{N}\log(1-D(s_E, a_E^i; \phi^i))\right]$$

The policy optimization in Eq. 2 could be viewed as finding a parameter $\theta$ that minimizes a surrogate objective of the log posterior conditioned on seeing $o_A = 1$, which implies to maximize the reward

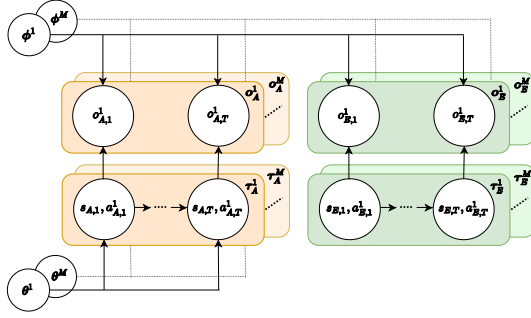 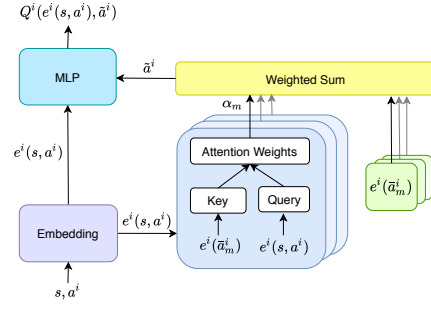

Figure 1: Probabilistic graphical model for the Bayesian multi-agent imitation learning

Figure 2: Attention mechanism for the mean field approximation

the agents received.

$$\log p(\boldsymbol{\theta} \mid \boldsymbol{o_A} = \boldsymbol{1}, \boldsymbol{\phi}) \propto \log \mathbb{E}_{\tau_A} \prod_{t=1}^{T} \prod_{i=1}^{N} p(o_{A,t}^i = 1 \mid s_{A,t}, a_{A,t}^i; \phi^i)$$

$$\geq \mathbb{E}_{\tau_A} \log \prod_{t=1}^{T} \prod_{i=1}^{N} p(o_{A,t}^i = 1 \mid s_{A,t}, a_{A,t}^i; \phi^i) = \mathbb{E}_{s_A, \boldsymbol{a}_A \sim \rho_{\pi_{\boldsymbol{\theta}}}} \left[ \sum_{i=1}^{N} \log D(s_A, a_A^i; \phi^i) \right] \quad (4)$$

### 3.1.2 Bayesian multi-agent imitation learning

Interacting with an environment to collect samples is an expensive operation in imitation learning. We seek to improve the sample efficiency through learning a discriminator that can produce more robust reward signals. To this end, instead of using a single point estimation to learn $\phi$ as in MA-GAIL, we propose to learn a distribution of the parameter $p(\phi)$. In this setting, one can derive a more robust estimation of the reward signals, which can better guide policy optimization.

To learn the optimal distribution of $\phi$, we use a prior distribution $p(\phi)$ as the initial distribution $q$, the surrogate objective of the posterior distribution $p(\phi \mid \boldsymbol{o_E} = \boldsymbol{0}, \boldsymbol{o_A} = \boldsymbol{1}, \boldsymbol{\theta})$ (Eq. 3) as the target distribution $p$, and minimize the distance between $p$ and $q$. To be specific, we adopted SVGD [14], where we draw $L$ samples of $\phi$ as an estimation of the distribution, and iteratively tune the samples to minimize the kernelized Stein discrepancy $S(q,p)$. The $l-th$ sample for agent $i$ can be updated as

$$\phi_l^i = \phi_l^i + \epsilon \varphi^i(\phi) \quad (5)$$

where $\epsilon$ is the step size, $\varphi^i(\phi) = \frac{1}{L} \sum_{l=1}^{L} \left( k(\phi_l^i, \phi) \delta_l^i + \nabla_{\phi_l^i} k(\phi_l^i, \phi) \right)$ is the updating gradient, $k$ is a positive definite kernel, and $\delta_l^i = \nabla_{\phi_l^i} \mathbb{E}_{\tau_E, \tau_A} \log \prod_{t=1}^{T} p(o_{E,t}^i = 0 \mid s_{E,t}, a_{E,t}^i; \phi_l^i) p(o_{A,t}^i = 1 \mid s_{A,t}, a_{A,t}^i; \phi_l^i)$ is the gradient from the surrogate loss for agent $i$.

For policy optimization, we optimize the surrogate objective of the posterior (Eq. 4) to find the best policy parameter $\boldsymbol{\theta}$, given a collected $L$ samples of the parameter $\phi$

$$\min_{\theta} \mathbb{E}_{\tau_A} \log \prod_{t=1}^{T} \prod_{i=1}^{N} p(o_{A,t}^i = 1 \mid s_{A,t}, a_{A,t}^i) = \min_{\theta} \sum_{i=1}^{N} \mathbb{E}_{s_A, a_A^i \sim \rho_{\pi_A^i}} \left[ \frac{1}{L} \sum_{l=1}^{L} \log D(s_A, a_A^i; \phi_l^i) \right]$$

where standard policy learning algorithms such as Q-learning [16] and actor-critic [13] methods could be applied. With this Bayesian formulation, existing multi-agent imitation learning algorithms could be reformulated in the Bayesian approach. For example, for MA-DAAC, if we use the same policy optimization method–the attention-actor-critic, but use multiple samples of the parameters $\phi$ with the updating method in Eq. 5, we derive a Bayesian MA-DAAC. Using the Bayesian formulation, we can generate more robust reward signals, increase the sampling efficiency, and improve the convergence rate, which we will demonstrate in the experiments section. More discussions about this Bayesian approach are presented in the Appendix. The remaining problem is how to improve the scalability so that the algorithm can still show prominent performance in large complex environments.

## 3.2 Mean field policy optimization

Existing approaches for multi-agent imitation learning have only been applied to scenarios with a small number of agents [7]. This is partly due to the fact that the more agents, the more information

each agent needs to take into consideration, and the more uncertainty and variance there is in policy search. To improve the scalability of the Bayesian multi-agent imitation learning, we introduce a multi-type mean field approximation into the policy search. We consider a system that categorizes agents into different types. Each agent type may contain different state and action spaces with different goals. The agents of the same type have the same state action spaces and reward functions. We introduce a new multi-type mean field approximation over the $Q$ value functions.

### 3.2.1 The mean field approximation

To simplify interactions among agents within one type, and to efficiently gather the information across types, we introduce a hierarchical mean field approximation. Consider a Markov game $G(\mathcal{S}, \mathcal{A}^1, \ldots, \mathcal{A}^N, r^1, \ldots, r^N, p, \gamma)$ with $M$ types of agents. Assume each agent belongs to one type, and each type contains $X_m$ agents. We use $a_m^{k_j}$ to denote the action for agent $k_j$ in type $m$. We assume $Q^i(s, \boldsymbol{a}) = Q^i(s, a^i, a_1^{k_1}, \ldots, a_1^{k_{X_1}}, \ldots, a_M^{k_1}, \ldots, a_M^{k_{X_M}})$ can be factorized into the weighted average of that only consider interactions within one type of agents $Q^i(s, a^i, a_m^{k_1}, \ldots, a_m^{k_{X_m}})$, which can be further decomposed additively into pairwise $Q^i(s, a^i, a_m^{k_j})$, i.e.,

$$Q^i(s, \boldsymbol{a}) = \sum_{m=1}^{M} \alpha_m Q^i(s, a^i, a_m^{k_1}, \ldots, a_m^{k_{X_m}}) = \sum_{m=1}^{M} \alpha_m \frac{1}{X_m} \sum_{j=1}^{X_m} Q^i(s, a^i, a_m^{k_j}) \tag{6}$$

Here $\alpha_m$ is the weight and $\sum_{m=1}^{M} \alpha_m = 1$. With this, the $Q$-function can be approximated with mean field theorem through Taylor approximation. Let $\bar{a}_m^i = \frac{1}{X_m} \sum_j a_m^{k_j}$ and $\tilde{a}^i = \sum_m \alpha_m \bar{a}_m^i$. We can show that

$$Q^i(s, \boldsymbol{a}) \approx Q^i(s, a^i, \tilde{a}^i) \tag{7}$$

The derivation of Eq. 7 is shown in Appendix. This hierarchical mean field approximation achieved using the weighted mean field to approximate the inter-types interactions, and vanilla mean field to approximate the intra-types interactions. This is intuitive because agents of the same type are similar to each other, and the interactions of which are capable to be estimated with a vanilla mean field. Agents of different types have more difference and we estimate with a weighted mean field which allows each agent to put different attention on different types of agents. This approximation on one hand significantly reduces the complexity of the interactions among agents and the variance of the $Q$ value function, on the other hand still preserves global interactions between any pair of agents implicitly.

**Theorem 3.1.** *Let the deviation of actions bounded by $\frac{1}{X_m} \sum_{j=1}^{X_m} \left| a_m^{k_j} - \tilde{a}^i \right| \leq \delta_m$ and $\sum_{m=1}^{M} \alpha_m \delta_m \leq \epsilon$, under the assumption given in Eq. 6, and that the Q function is $K - Lipschitz$, then the error of the multi-type mean field Q function is bounded by*

$$|Q^i(s, \boldsymbol{a}) - Q^i(s, a^i, \tilde{a}^i)| \leq K\epsilon$$

With Theorem 3.1, we bound the error of our multi-type mean field approximation. In the following, we will show how to learn this $Q$-function with multi-type mean field approximation.

### 3.2.2 Mean field update

The Q function for agent $i$ can be updated iteratively with TD method

$$Q^i(s, a^i, \tilde{a}^i) = (1 - \alpha)Q^i(s, a^i, \tilde{a}^i) + \alpha \left( r^i + \gamma V^i(s') \right) \tag{8}$$

In the above, the reward is estimated through the learned parameters $\phi_l^i$ in Eq. 5

$$r^i = -\frac{1}{L} \sum_{l=1}^{L} \log D(s, a^i; \phi_l^i) \tag{9}$$

The value function can be estimated as

$$V^i(s) = \sum_{a^i} \pi_i(a^i \mid s, \tilde{a}^i) \mathbb{E}_{a^{-i} \sim \pi^{-i}} Q^i(s, a^i, \tilde{a}^i) \tag{10}$$

where the multi-type mean field action

$$\tilde{a}^i = \sum_m \alpha_m \bar{a}^i_m, \qquad \bar{a}^i_m = \frac{1}{X_m} \sum_{j=1}^{X_m} a^{k_j}_m, \qquad a^{k_j}_m \sim \pi(\cdot \mid s, \tilde{a}^{k_j}_m) \qquad (11)$$

The action for each agent $a^i$ is chosen following a Boltzmann policy

$$\pi(a^i \mid s, \tilde{a}^i) = \frac{\exp(\beta Q^i(s, a^i, \tilde{a}^i))}{\sum_{a^{i'}} \exp(\beta Q^i(s, a^{i'}, \tilde{a}^i))} \qquad (12)$$

Alternatively, we can estimate the policy explicitly with a neural network, the gradient of which can be computed as

$$\nabla_{\theta^i} \log \pi_{\theta^i}(s) Q^i(s, a^i, \tilde{a}^i) \mid_{a^i = \pi_{\theta^i}(s)} \qquad (13)$$

**Theorem 3.2.** *Under assumptions 1). Each action-value pair for multi-type mean field settings is visited infinitely often, and the reward is bounded by some constant. 2). Agents policies are Greedy in the Limit with Infinite Exploration (GLIE). In the case with the Boltzmann policy, the policy becomes greedy w.r.t. the Q-function in the limit as the temperature decays asymptotically to zero. 3). For each stage game $[Q^1_t(s), \dots, Q^N_t(s)]$ at time $t$ and in state $s$ in training, for all $t$, $s$, $i \in \{1, \dots, N\}$, the Nash equilibrium $\pi_* = [\pi^1_*, \dots, \pi^N_*]$ is recognized either as a). the global optimum or b). a saddle point expressed as $\mathbb{E}_{\pi_*}[Q^i_t(s)] \geq \mathbb{E}_{\pi}[Q^i_t(s)]$, $\forall \pi \in \Omega(\Pi_k \mathcal{A}^k)$; or $\mathbb{E}_{\pi_*}[Q^i_t(s)] \geq \mathbb{E}_{\pi^i} \mathbb{E}_{\pi^{-i}_*}[Q^i_t(s)]$, $\forall \pi^i \in \Omega(\mathcal{A}^i)$ and $\mathbb{E}_{\pi_*}[Q^i_t(s)] \leq \mathbb{E}_{\pi^i_*} \mathbb{E}_{\pi^{-i}}[Q^i_t(s)]$, $\forall \boldsymbol{\pi}^{-i} \in \Omega(\Pi_{k \neq i} \mathcal{A}^k)$.*

*If we update the multi-type mean field approximation Q value function $Q^i(s, a^i, \tilde{a}^i)$ for each agent according to Eq. 8, 10, 11 and 12, it will converge to a Nash-Q value $\boldsymbol{Q}^*$ with error bounded by $K\epsilon - S$, where $S$ is a constant, $K$ and $\epsilon$ are from Theorem 3.1.*

Theorem 3.2 follows the same structure as Theorem 3.4 in [3], but differs in that we introduce a new multi-type mean field approximation $Q^i(s, a^i, \tilde{a}^i)$. It shows that our approach converges to a fixed point within a small bounded distance of the Nash equilibrium over the learned reward function.

### 3.2.3 An attention mechanism for the mean field approximation

In practice, we perform state-action embeddings $e^i(s, a^i)$ and mean-field action embeddings $e^i_m(\bar{a}^i_m)$ before sending $(s, a^i, \tilde{a}^i)$ to the $Q$ value function, which transforms different action dimensions of each types, if any, to an embedding space with the same dimensions. The Q function with embeddings can be represented as $Q^i(e^i(s, a^i), \tilde{a}^i)$, where $\tilde{a}^i = \sum_m \alpha_m e^i_m(\bar{a}^i_m)$.

We implement the hierarchical mean field approximation of the Q function $Q^i(e^i(s, a^i), \tilde{a}^i)$ through an attention mechanism, as illustrated in Figure 2. The mean field within each type $\bar{a}^i_m$ can be computed through averaging the actions of each agent (element-wise average of an action vector, for both continuous and discrete action spaces). The weighted mean field between each type, $\tilde{a}^i = \sum_m \alpha_m e^i_m(\bar{a}^i_m)$, can be viewed as each agent querying each type for the information of their mean actions, where the weight $\alpha_m$ reflects the attention that this agent puts on each types.

The mean field weight $\alpha_m$ is implemented as the attention weight comparing the state action embedding $e^i(s, a^i)$ and the mean field action embedding $e^i_m(\bar{a}^i_m)$ in terms of the dot similarity, which is evaluated through a softmax function, *i.e.*,

$$\alpha_m \propto \exp\left(e^i_m(\bar{a}^i_m)^T W^T_k W_q e^i(s, a^i)\right) ,$$

where $W_q$ transforms $e^i(s, a^i)$ into a "query" and $W_k$ transforms $e^i_m(\bar{a}^i_m)$ into a "key". The weights $W_k$ and $W_q$ are shared among agents, which allows learning more efficiently as the aggregating of information for each agents are different but share some common features.

### 3.3 Bayesian multi-type mean field multi-agent imitation learning

In this section, we present our algorithm, Bayesian multi-type mean field multi-agent imitation learning (BM3IL). We optimize with respect to a Bayesian framework of multi-agent imitation learning, meanwhile using a multi-type mean field approximation in policy optimization. The

objective function is given below, where $\rho_{\pi_A^i(\cdot|s,\tilde{a}^i;\theta^i)}$ indicates the samples are generated using the policy with the mean field approximation.

$$\min_\theta \max_\phi \sum_{i=1}^N \mathbb{E}_{s,a^i \sim \rho_{\pi_A^i(\cdot|s,\tilde{a}^i;\theta^i)}} \left[ \frac{1}{L} \sum_{l=1}^L \log D(s,a^i;\phi_l^i) \right] + \sum_{i=1}^N \mathbb{E}_{s,a^i \sim \rho_{\pi_E^i}} \left[ \frac{1}{L} \sum_{l=1}^L \log \left( 1 - D(s,a^i;\phi_l^i) \right) \right]$$

We iterate between optimizing the reward function and optimizing the policy. For reward learning, we use SVGD to obtain $L$ point estimation of the distribution of parameter $\phi$. For policy optimization, we use an actor-critic method where we estimate the $Q$ value with attention networks and model the policy explicitly with fully connected neural networks. The algorithm is given in Algorithm 1.

---

**Algorithm 1:** Bayesian multi-type mean field multi-agent imitation learning

---

**Input:** A Markov game, expert dataset of trajectories $\{\{s_{E,t}^i, a_{E,t}^i\}_{i=1}^N\}_{t=1}^T$, initial policy
  parameters $\boldsymbol{\theta} = \{\theta^i\}_{i=1}^N$, $L$ samples of the reward function parameters $\boldsymbol{\phi} = \{\{\phi_l^i\}_{i=1}^N\}_{l=1}^L$
  sampled from the prior distribution $p(\boldsymbol{\phi})$, and Q value function parameters $\boldsymbol{\omega} = \{\omega^i\}_{i=1}^N$
**Output:** Learned policy parameters $\boldsymbol{\theta}$
for $iter = 0, 1, ...$ do
  Sample trajectories $\{\{s_{A,t}^i, a_{A,t}^i\}_{i=1}^N\}_{t=1}^M$ by executing $\pi_\theta$
  Update reward parameters $\boldsymbol{\phi}$ with Eq. 5
  Estimate the reward with Eq. 9
  Update the Q parameters $\boldsymbol{\omega}$ through Eq. 8
  Update policy parameters $\boldsymbol{\theta}$ with gradient in Eq. 13
end

---

## 4  Connections with existing works

[3] developed a multi-type mean field approximation to solve a reinforcement learning problem, where they assume all types having the same amount of agents, and they only apply mean field for agents within each type. Comparing to their work, we develop a new multi-type mean field approximation which does not require the number of agents for each type being equal, and which approximates both inter-types and intra-types interactions among agents. By allowing agents to put different attention on different types, our method provides more effective information gathering.

Our work is partly inspired by [8], which applied Bayesian approach to single-agent imitation learning. But they do not consider the discounted factor $\gamma$, nor do they give the meaning of the observations. We derive a more complete formulation of the Bayesian approach for multi-agent imitation learning.

[18] developed a multi-agent generative adversarial imitation learning (MA-GAIL) through extending the single-agent generative adversarial imitation learning, and [7] developed a multi-agent discriminator-actor-attention-critic (MA-DAAC) which improves the MA-GAIL through integrating attention-actor-critic in the policy optimization. Comparing to their versions, first, we developed a Bayesian multi-agent imitation learning framework which learn a distribution of the reward parameters rather than a point estimation, and which improves the sample efficiency. Second, in policy optimization, MA-GAIL does not consider other agents' actions. MA-DAAC considers every agent's actions with an actor-attention-critic, which is not scalable. We introduce a multi-type mean field to capture the interactions with other agents, which improves the scalability.

## 5  Experiments

In this section, we demonstrate the power of our algorithm empirically from the following perspectives: (1) How the Bayesian approach improves the sampling efficiency of existing multi-agent imitation learning. (2) How our algorithm, BM3IL, improves the scalability and the sampling efficiency. The benchmarking algorithms include MA-GAIL [18], MA-DAAC [7] and MA-GAIL incorporating the existing multi-type mean field approximation [3] in the policy optimization, which we call MTMFIL. For fairness, all algorithms implement the same neural network structure. Details and more experiments are presented in the Appendix.

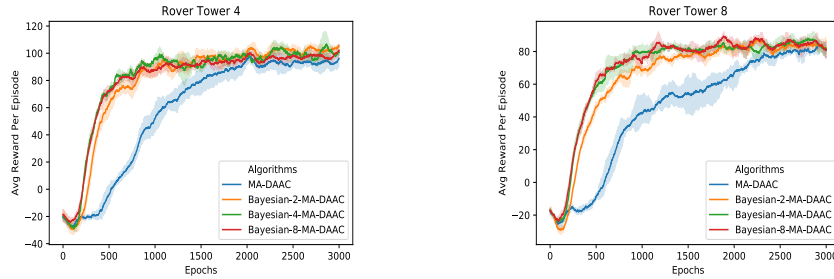

Figure 3: Learning curves of each algorithm in the Rover Tower environment. Left: 4 agents. Right: 8 agents. For the Bayesian algorithms, we vary the number of samples of the parameters $\phi$, as indicated by the number after 'Bayesian'. For example, Bayesian-2-MA-DAAC means using 2 samples of the parameters $\phi$ for each agents.

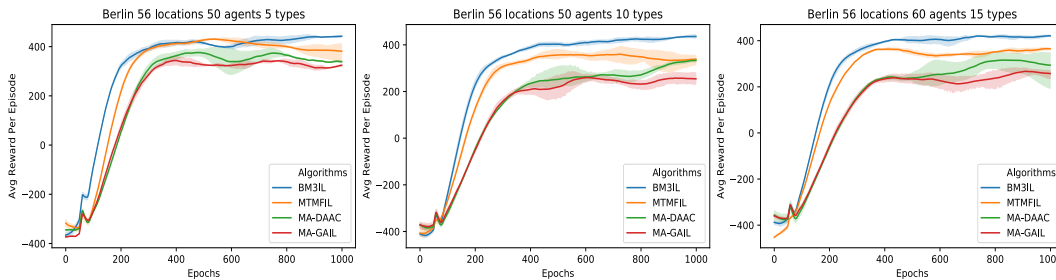

Figure 4: Learning curves of each algorithm in the transportation environment

## 5.1 Sample efficiency of the Bayesian approach

To demonstrate how the Bayesian framework improves the sample efficiency, we compare the performance of MA-DAAC with a Bayesian MA-DAAC, which uses the same algorithm as MA-DAAC except using multiple discriminator networks to represent the multiple samples of the parameter $\phi$. Noted that in this experiment, the goal is to demonstrate the contribution of the Bayesian formulation, hence we do not apply the multi-type mean field approximation. The environment we used in this experiment is the Rover Tower [6]. It involves a total of 4 or 8 agents, of which half are 'rovers' and half are 'towers'. The goal is to let the towers to navigate the rovers to arrive at their destinations. We vary the total number of agents and the number of samples of the parameters $\phi$ (the number of discriminator networks for each agent).

The learning curves are shown in Figure 3. The experimental results showed that the Bayesian approach converges faster than the MA-DAAC. Using 2 samples of the $\phi$ already accelerates the convergent speed significantly. Further improve the number of samples slightly improve the convergent speed. The Bayesian approach outperforms the single point estimation counterpart MA-DAAC because it learns a distribution of the discriminator, which provides more robust reward signals to better guide the learning of the policy.

## 5.2 Performance of BM3IL in complex environment

To test the scalability and sampling efficiency of our method with the mean field approximation, we benchmark BM3IL with MA-GAIL, MA-DAAC, and MTMFIL in a transportation environment with a real-world network. The environment we used is Berlin, which contains 46 road locations, 10 facilities, and 50 agents representing 5000 vehicles that are supposed to go to work in the morning and go back home at night, where each agent represent 100 vehicles. The agents are categorized into different types, each type having their own home and work facilities. We vary the number of agents (50, 60) and the number of types (5, 10, 15) in this environment.

The learning curves are shown in Figure 4. The experimental results showed that BM3IL outperforms all other algorithms with faster convergence speed. Moreover, the more complex the environment (as

the more types and more agents), the more performance gain it achieves. MA-GAIL performs worst because it does not consider other agents' actions. MA-DAAC gathers other agents' information through an actor-attention-critic, which performs better than MA-GAIL, but which is not as efficient and scalable as our method. MTMFIL uses a multi-type mean field approximation which can not put different attention on different types. Our algorithm achieved the best performance by using the Bayesian approach to stabilize the reward signals, and the multi-type mean field approximation to efficiently gather the information and improves scalability.

## 6   Conclusion

In this paper, we developed BM3IL, a multi-agent imitation learning algorithm. Our algorithm improves sample efficiency by using a Bayesian formulation and improves scalability by introducing a new multi-type mean field approximation. The benchmarking with 3 state-of-the-art algorithms indicated that our method achieved the best sample efficiency and scalability.

## 7   Broader impact

From the research perspective, our work is the first to connect the attention mechanism with mean field approximation, which connects the neural network community with the game theory community. On the other hand, our work is also the first to introduce the concept of mean field to the multi-agent imitation learning, which brings the game theory community into the imitation learning community. The combination of the communities points out new research directions where more research opportunities may be found.

From the society perspective, first, multi-agent imitation learning has a large number of real-world applications. We live in a world full of complex multi-agent systems, such as the transportation system where each vehicle or each group of vehicles can be viewed as an agent, or the epidemic system where each individual or each group of people can be viewed as an agent. Optimizing the policy in these real-world multi-agent systems are important and valuable, such as optimizing the driving route of each vehicle to reduce the driving time, or optimizing the medical resource allocation and regulating the interaction of people to mitigate the spread of epidemic disease. Reward function is not given in such real-world systems. Instead, we may have expert demonstrations that we can mimic, such as how experienced drivers drive in a transportation system. As such, optimizing the policies in real-world multi-agent systems can be formulated as multi-agent imitation learning problems.

Second, our work has the potential to solve real-world multi-agent imitation learning problems. MAIL is still a relatively new domain. The challenges for applying MAIL to solve real-world problems are scalability and sampling efficiency since interactively collecting the samples is an expensive operation and real-world systems often have a large number of agents. Our work is one step in this direction. In this paper, we improve the scalability by introducing the Bayesian formulation, and improve the sample efficiency through introducing a new multi-type mean filed approximation, as demonstrated in the experiments. Our work has the potential to solve real-world MAIL problems.

The limitation of our approach is still scalability. Currently, we can not scale to environments with thousands of agents and hundreds of types where the state-action space has thousands of dimensions. Further improve the scalability would be the future work.

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
