[Supplementary Material 1 · appendix.pdf]

# 8 Appendix

## 8.1 More discussions about the Bayesian multi-agent imitation learning

In this paper, for the Bayesian framework, we learn the reward function and optimize the policy by optimizing the surrogate objective of the posterior distribution as in Eq. 3 and 4. Alternatively, we can directly use Bayesian inference to optimize the true posterior of the probabilistic graphical model instead of the surrogate objective of the posterior.

Overall, inference the true posterior in a probabilistic graphical model which is defined in Figure 1 does not exactly optimize the same objective function as a standard imitation learning as in Eq. 1 and 2. This is also pointed out in other papers connecting variational inference with reinforcement learning [12]. As such, inference the true posterior should provide comparable result, but does not optimize the true objective function as a standard imitation learning.

Sample efficiency is a key issue in multi-agent imitation learning, and collecting samples in the real-world (such as a transportation system) is expensive. Bayesian parameter estimator can take full account of the uncertainties related the cost-function parameters $\phi$ compared to a point estimator, and is shown in Section 5.1 to improve sample efficiency by enhancing exploration.

## 8.2 Derivation of equations

Derivation of Eq. 7

$$
\begin{aligned}
Q^i(s, \boldsymbol{a}) &= \sum_{m=1}^{M} \alpha_m \left( \frac{1}{X_m} \sum_{j=1}^{X_m} Q^i(s, a^i, a_m^{k_j}) \right) \\
&\approx \sum_{m=1}^{M} \alpha_m \frac{1}{X_m} \sum_{j=1}^{X_m} \left( Q^i(s, a^i, \bar{a}_m^j) + \nabla_{\bar{a}_m^i} Q^i(s, a^i, \bar{a}_m^i)(a_m^{k_j} - \bar{a}_m^i) \right) \\
&= \sum_{m=1}^{M} \alpha_m Q^i(s, a^i, \bar{a}_m^i) \\
&\approx \sum_{m=1}^{M} \alpha_m \left( Q^i(s, a^i, \tilde{a}^i) + \nabla_{\tilde{a}^i} Q^j(s, a^i, \tilde{a}^i)(\bar{a}_m^i - \tilde{a}^i) \right) \\
&= Q^i(s, a^i, \tilde{a}^i)
\end{aligned}
$$

## 8.3 Proof of theorems

*Proof.* of Theorem 3.1

$$
\begin{aligned}
\left| Q^i(s, \boldsymbol{a}) - Q^i(s, a^i, \tilde{a}^i) \right| &= \left| \sum_{m=1}^{M} \alpha_m \frac{1}{X_m} \sum_{j=1}^{X_m} Q^i(s, a^i, a_m^{k_j}) - Q^i(s, a^i, \tilde{a}^i) \right| \\
&= \left| \sum_{m=1}^{M} \alpha_m \frac{1}{X_m} \sum_{j=1}^{X_m} Q^i(s, a^i, a_m^{k_j}) - \sum_{m=1}^{M} \alpha_m \frac{1}{X_m} \sum_{j=1}^{X_m} Q^i(s, a^i, \tilde{a}^i) \right| \\
&\leq \sum_{m=1}^{M} \alpha_m \frac{1}{X_m} \sum_{j=1}^{X_m} \left| Q^i(s, a^i, a_m^{k_j}) - Q^i(s, a^i, \tilde{a}^i) \right| \\
&\leq \sum_{m=1}^{M} \alpha_m \frac{1}{X_m} \sum_{j=1}^{X_m} K \left| a_m^{k_j} - \tilde{a}^i \right| \\
&\leq K\epsilon
\end{aligned}
$$

Where the 2nd step is due to $\sum_{m=1}^{M} \alpha_m = 1$. The 3rd step is due to the triangle inequality. The 4th step is due to that the Q function is $K - $ Lipschitz.

$\square$

In order to proof Theorem 3.2, we first introduce Lemmas 8.1 and 8.2.

**Lemma 8.1.** *Under the assumption 3 in Theorem 3.2, the Nash operator $\mathscr{H}^{Nash}$ forms a contraction mapping on the complete metric space from $\mathscr{Q}$ to $\mathscr{Q}$ with the fixed point being the Nash Q-value of the entire game, i.e. $\mathscr{H}^{Nash}\boldsymbol{Q}_* = \boldsymbol{Q}_*$.*

*Proof.* Please refer to Theorem 17 in [5] for a detailed proof. $\square$

**Lemma 8.2.** *The random process $\{\Delta_t\}$ defined in $\mathbb{R}$ as*

$$
\Delta_{t+1}(x) = (1 - \alpha_t(x))\Delta_t(x) + \alpha_t(x)F_t(x) \tag{14}
$$

*converges to a constant $S$ with probability 1 (w.p.1) when*

*1.* $0 \leq \alpha_t(a) \leq 1, \sum_t \alpha_t(x) = \infty, \sum_t \alpha_t^2 < \infty$;

*2.* $x \in \mathcal{X}$, *the set of possible states, and* $|\mathcal{X}| < \infty$;

*3.* $\|\mathbb{E}[F_t(x)|\mathscr{F}_t]\|_W \leq \gamma\|\Delta_t\|_W + R$, *where* $\gamma \in [0, 1)$ *and* $R$ *is finite;*

*4.* $\boldsymbol{var}[F_t(x)|\mathscr{F}_t] \leq R_2(1 + \|\Delta_t\|_W^2)$ *with constant* $R_2 > 0$.

*Here* $\mathscr{F}_t$ *denotes the filtration of an increasing sequence of* $\sigma$*-fields including the history of processes;* $\alpha_t, \Delta_t, F_t \in \mathscr{F}_t$ *and* $\|\cdot\|_W$ *is a weighted maximum norm. The value of this constant* $S = \frac{\psi C_1 + \alpha |R|}{\alpha \beta_0}$ *where* $\psi \in (0, 1)$ *and* $C_1$ *is the value with which the scale invariant iterative process is bounded,* $\beta_0$ *is the scale factor applied to the original process.*

*Proof.* Please refer to Appendix C in [3] for a detailed proof. $\square$

*Proof.* of Theorem 3.2 is following the structure of Theorem 3.4 in [3]. The difference is that we introduce a new mean field approximation. We outline the proofs using our notations below.

We define $\boldsymbol{Q}^M$ as the concatenation of $[Q^1(s, a^1, \tilde{a}^1), \dots, Q^N(s, a^N, \tilde{a}^N)]$. By subtracting $\boldsymbol{Q}_*(s, \boldsymbol{a})$ on both sides of Eq. 8 and in relation to Eq. 14:

$$\Delta_t(x) = \boldsymbol{Q}_t^M - \boldsymbol{Q}_*(s, \boldsymbol{a}) \tag{15}$$
$$\boldsymbol{F}_t(x) = \boldsymbol{r}_t + \gamma \boldsymbol{V}_t^{MF}(s_{t+1}) - \boldsymbol{Q}_*(s_t, \boldsymbol{a}_t)$$

where $x \triangleq (s_t, \boldsymbol{a}_t)$ denotes the visited state-action pair at time $t$. In Eq. 14, $\alpha_t$ is interpreted as the learning rate with $\alpha_t(s', \boldsymbol{a}') = 0$ for any $(s', \boldsymbol{a}') \neq (s_t, \boldsymbol{a}_t)$; this is because that each agent only updates the Q-function with the state stand actions $\boldsymbol{a}_t$ visited at time $t$. Hence, the first condition of Lemma 8.2 is automatically satisfied. The second condition also holds as we are considering finite state and action spaces.

In Theorem 3.1, we showed a bound for the actual Q function and the multi type mean field Q function. We apply that in Eq. 15, to get the following equation for $\Delta$.

$$\begin{aligned}
\Delta_t(x) &= \boldsymbol{Q}_t^M - \boldsymbol{Q}_*(s, \boldsymbol{a}) \\
&= \boldsymbol{Q}_t^M + \boldsymbol{Q}_t(s, \boldsymbol{a}) - \boldsymbol{Q}_t(s, \boldsymbol{a}) - \boldsymbol{Q}_*(s, \boldsymbol{a}) \\
&\leq |\boldsymbol{Q}_t^M - \boldsymbol{Q}_t(s, \boldsymbol{a})| + \boldsymbol{Q}_t(s, \boldsymbol{a})) - \boldsymbol{Q}_*(s, \boldsymbol{a}) \\
&\leq \boldsymbol{Q}_t(s, \boldsymbol{a})) - \boldsymbol{Q}_*(s, \boldsymbol{a}) + D
\end{aligned} \tag{16}$$

where $D = K\epsilon$ is from Theorem 3.1.

We need to show that the mean field multi type operator $\mathscr{H}^{MTMF}$ meets Lemma 8.2 third and fourth conditions and that $\Delta$ in Eq. 16 converges to a constant S according to Lemma 8.2.

Let $\mathscr{F}_t$ denote the $\sigma$-field generated by all random variables in the history time $t - (s_t, \alpha_t, a_t, r_{t-1}, \dots, s_1, \alpha_1, \boldsymbol{a_1}, \boldsymbol{Q}_0)$. Thus, $\boldsymbol{Q}_t$ is a random variable derived from the historical trajectory up to time $t$. Given the fact that all $\boldsymbol{Q}_\tau$ with $\tau < t$ are $\mathscr{F}_t$-measurable, both $\Delta_t$ and $\boldsymbol{F}_{t-1}$ are therefore also $\mathscr{F}_t$-measurable, which satisfies the measurability condition of Lemma 8.2.

To prove the third condition of Lemma 8.2 we begin with Eq. 15 that

$$\begin{aligned}
\boldsymbol{F}_t(s_t, \boldsymbol{a}_t) &= \boldsymbol{r}_t + \gamma \boldsymbol{V}_t^{\text{MTMF}}(s_{t+1}) - \boldsymbol{Q}_*(s_t, \boldsymbol{a}_t) \\
&= \boldsymbol{r}_t + \gamma \boldsymbol{V}_t^{\text{Nash}}(s_{t+1}) - \boldsymbol{Q}_*(s_t, \boldsymbol{a}_t) + \gamma[\boldsymbol{V}_t^{\text{MTMF}}(s_{t+1}) - \boldsymbol{V}_t^{\text{Nash}}(s_{t+1})] \\
&= [\boldsymbol{r}_t + \gamma \boldsymbol{V}_t^{\text{Nash}}(s_{t+1}) - \boldsymbol{Q}_*(s_t, \boldsymbol{a}_t)] + C_t(s_t, \boldsymbol{a}_t) \\
&= \boldsymbol{F}_t^{\text{Nash}}(s_t, \boldsymbol{a}_t) + C_t(s_t, \boldsymbol{a}_t)
\end{aligned} \tag{17}$$

$\boldsymbol{F}_t^{Nash}$ in Eq. 17 is the same as $\boldsymbol{F}_t$ in Lemma 8.2 in proving the convergence of the Nash Q-learning algorithm. From Lemma 8.1, $\boldsymbol{F}_t^{Nash}$ forms a contraction mapping with the norm $||\cdot||_\infty$ being the maximum norm on $\boldsymbol{a}$. Thus from Eq. 16,

$$
\begin{aligned}
|| \mathbb{E}[\boldsymbol{F}_t^{Nash}(s_t, a_t)|\mathscr{F}_t]||_\infty &\leq \gamma||\boldsymbol{Q}_* - \boldsymbol{Q}_t||_\infty \leq \gamma||D - \Delta_t||_\infty \\
&= ||F_t^{Nash}(s_t, \boldsymbol{a}_t)|\mathscr{F}_t||_\infty + ||C_t(s_t, \boldsymbol{a}_t)|\mathscr{F}_t||_\infty \\
&\leq \gamma||D - \Delta_t||_\infty + ||C_t(s_t, \boldsymbol{a}_t)|\mathscr{F}_t||_\infty \\
&\leq \gamma||\Delta_t||_\infty + ||C_t(s_t, \boldsymbol{a}_t)|\mathscr{F}_t||_\infty + \gamma||D||_\infty \quad (18)\\
&\leq \gamma||\Delta_t||_\infty + \gamma|D| \quad (19)
\end{aligned}
$$

In Eq. 18, two last terms are both positive and finite. It can be proved that the term $||C_t(s_t, \boldsymbol{a}_t)||_\infty$ converges to zero $w.p.$ 1. Please refer to Theorem 1 in [21] for more details. Thus, the third condition of Lemma 8.2 is proved. The value of constant $R = \gamma|D| = \gamma|K\epsilon|$.

In Lemma 8.2 for the fourth condition we use the fact that the MTMF operator $\mathscr{H}^{MTMF}$ forms a contraction mapping, $i.e.$ $\mathscr{H}^{MTMF}\boldsymbol{Q}_* = \boldsymbol{Q}_*$ and it follows that:

$$
\begin{aligned}
\mathbf{var}[\boldsymbol{F}_t(s_t, \boldsymbol{a}_t)|\mathscr{F}_t] &= \mathbb{E}[(\boldsymbol{r}_t + \gamma\boldsymbol{V}_t^{MTMF}(s_{t+1}) - \boldsymbol{Q}_*(s_t, \boldsymbol{a}_t))^2] \\
&= \mathbb{E}[(\boldsymbol{r}_t + \gamma\boldsymbol{V}_t^{MTMF}(s_{t+1}) - \mathscr{H}^{MTMF}(\boldsymbol{Q}_*))^2] \\
&= \mathbf{var}[\boldsymbol{r}_t + \gamma\boldsymbol{V}_t^{MTMF}(s_{t+1})|\mathscr{F}_t] \leq R_2(1 + ||\Delta_t||_W^2) \quad (20)
\end{aligned}
$$

In Eq. 20 the reward is bounded by some constant, employed from Assumption 1 and the value function is also bounded by being updated recursively by Eq. 10. So we can choose a positive, finite $R_2$ such that the inequality holds. Finally, with all conditions met, it follows from Lemma 8.2 that $\Delta_t$ converges to constant S with probability 1, where $S = \frac{\psi C_1 + \alpha\gamma|D|}{\alpha\beta_0}$ from Lemma 2 and using the value of $R_2$ derived above. Therefore, from Eq. 16 we get:

$$
\boldsymbol{Q}_*(s, \boldsymbol{a}) - \boldsymbol{Q}_t(s, \boldsymbol{a}) \leq D - S \leq K\epsilon - S
$$

$\square$

## 8.4 More experimental details and further experimental results

The experiments were run on the server with the following characteristics: Intel Xeon Gold 6230 (2/node), NVidia Tesla V100 24GB (2/node).

**Environments**  Rover Tower environment originates from [6]. Main goal is the interaction between the randomly paired agents of "Towers" and "Rover", where "Tower" communicate with "Rovers" to arrive at their destination. Berlin transportation environment [19] is based on the MATSim open Berlin scenario.

**Sample efficiency of the Bayesian approach**  In the Rover Tower environment, for the Rover, the state dimension is 11, and the action dimension is 5. For the Tower, the state dimension is 6 and the action dimension is 5. The expert demonstrations are collected through running the MA-DAAC algorithm until convergent with the associated reward function of this environment. Then we run imitation learning algorithms with 300 trajectories of the expert data. We run each algorithms 6 times and plotted the mean and standard deviation (shaded area) of the results as in Figure 3. The statistics of the Rover Tower experiments are summarized in Table 1.

**Experiment on more environment**  Figure 5. shows the performance comparison in Cooperative communication environment (Multi-Agent Actor-Critic for Mixed Cooperative-Competitive Environments (Ryan et al., 2017))

| Environment | Rover Tower 4 | Rover Tower 8 |
|---|---|---|
| Expert | **133.20 ± 57.43** | **113.70 ± 50.25** |
| Bayesian-2-MA-DAAC | **134.40 ± 25.78** | 109.17 ± 34.90 |
| Bayesian-4-MA-DAAC | 118.16 ± 26.19 | 98.60 ± 37.05 |
| Bayesian-8-MA-DAAC | 123.41 ± 13.52 | 96.17 ± 33.66 |
| MA-DAAC | 121.80 ± 24.24 | 101.26 ± 35.92 |

Table 1: The mean and standard deviation of the reward of the learned policy for the Rover Tower environment

| Environment | Berlin (50,5) | Berlin (50,10) | Berlin (60,15) |
|---|---|---|---|
| Expert | **506.47 ± 16.21** | **481.69 ± 16.71** | **472.12 ± 19.87** |
| BM3IL | **450.36 ± 137.97** | **437.68 ± 115.22** | **421.03 ± 136.85** |
| MA-GAIL | 360.56 ± 211.28 | 283.47 ± 256.26 | 303.09 ± 233.34 |
| MA-DAAC | 389.65 ± 195.14 | 316.00 ± 225.17 | 334.00 ± 217.77 |
| MTMFIL | 338.75 ± 189.98 | 321.75 ± 225.41 | 339.47 ± 207.78 |

Table 2: The mean and standard deviation of the reward of the learned policy for the transportation environment, where the bracket means (the number of agents, the number of types)

Figure 5: The learning curve for Cooperative communication environment

**Performance of BM3IL in complex environment**     In the Berlin environment, for each agent, the state dimension is 66, which includes the agent location of each agent plus the goal destination in categorical distribution. The action dimension is 56, which is a categorical distribution indicating the next location that this agent will move to. We implement the action with a masked network such that at each location, the valid action space is only a subset of all locations. The expert demonstration is collected through running the MA-DAAC algorithm until convergent with the associated reward function of this environment. Then we run imitation learning algorithms with 70 trajectories of the expert data. We run each algorithms 6 times and plotted the mean and standard deviation (shaded area) of the results as in Figure 4.The statistics of the transportation experiments are summarized in Table 2.

To have a straightforward view of the performance of each algorithm, for Berlin environment with 50 agents and 10 types, we draw the number of agents at their goal location at each time step for each types using the learned policy for each algorithm, as shown in Figure 6. As we can see, BM3IL achieves the most amount of agents being at the goal locations for most of the time, comparing to MA-GAIL, MA-DAAC, and MTMFIL.

Figure 6: The number of agents at the goal locations for each type for each algorithms in the environment Berlin type 10



[Supplementary Material 2]



Berlin 56 locations 60 agents 15 types

Avg Reward Per Episode vs Epochs

Algorithms:
- BM3IL
- MTMFIL
- MA-DAAC
- MA-GAIL

[Supplementary Material 3 · berlin-type5.pdf]



Berlin 56 locations 50 agents 5 types

Avg Reward Per Episode vs Epochs

Algorithms
- BM3IL
- MTMFIL
- MA-DAAC
- MA-GAIL