[Reviews · NeurIPS 2020]

Review 1

Summary and Contributions: This paper proposes to improve multi-agent imitation learning with two enhancements: 1) learn a distribution over phi, the parameter of the discriminator used to estimate the reward function, 2) improve on the multi-type mean field approximation for multi-agent RL by allowing a different number of agents per type. The results show that this leads to faster learning in Rover Tower and transportation network tasks.

Strengths: The empirical evaluation of this paper is strong, in that it benchmarks against 3 relevant and recent baselines for multi-agent imitation learning, and clearly exceeds them. The paper provides theoretical grounding in the mean field approximation and its convergence.

Weaknesses: A weakness of the paper is the novelty of this approach above prior work; Section 4 states that [3] also used a multi-agent multi-type mean field approximation, only it did not allow types to have a different number of agents. Is this the only difference from this prior work? This should be clarified.

Correctness: I did not notice any major errors. Figures 3 & 4 should include a description of how the error bars were calculated.

Clarity: The paper is reasonably clear, but there are some typos and errors. For example, the sentence describing Rover Tower in line 250, "The goal is to let the towers to navigate the rovers to arrive at their destinations" is extremely unclear, and interferes with understanding the experiment. It would also help to provide a citation or reference to the Berlin transportation environment. In addition, I felt that the clarity of the paper could be improved by devoting more explanation to the new mean field formulation, why/how it is different from prior work, and why some of the assumptions are justified. For example, why is it reasonable to assume that it is only necessary to model interactions within one type of agent (lines 151-152)? To save space for this discussion, well-known equations (e.g. 8 and 12) could be removed.

Relation to Prior Work: As mentioned above, the distinction between this paper and [3] should be made more clear. Lines 232-233 state that MA-DAAC is not scalable because it uses attention to consider every agent's actions. Why is this not scalable? Attention can summarize a variable number of agents' actions in a meaningful way, and attention is often promoted as a mechanism for dealing with increased numbers of agents.

Reproducibility: Yes

Additional Feedback: - The contributions section (lines 41-46) appears to re-state the same contribution several times in different ways. Perhaps if each contribution were made more concise, the differences would be more clear. - Line 66: "Nash equilibrium described" -> describes - Line 189: "we introduces" -> we introduce ** I have read the authors' rebuttal and appreciated the clarifications provided, so I am not changing my review score. I hope the authors will work these clarifications into the final version of the paper.


Review 2

Summary and Contributions: The paper propose to improve the sample efficiency and scalability of MA-GAIL algorithm by introducing a Bayesian formulation as well as a new multi-type mean field approximation. Some theoretical analysis on the proposed multi-type mean field approximation was provided and experimental results on some simulated environments are provided to justify the above claims.

Strengths: The paper contains rich discussions on preliminaries and detailed derivations of the proposed algorithm. The proposed multi-type mean-filed approximation seems to have the potential of improving the scalability of existing multi-agent reinforcement learning algorithms. The authors also made efforts to analyze the approximation theoretically and show that the approximation error will not lead to large deviations from optimal solutions under certain conditions.

Weaknesses: - I think this paper is a natural extension/combination of previous works [1] and [2], especially the bayesian formulation and the resulting algorithm. And the paper does not give enough insights/analysis of why such a bayesian framework provides special advantages in the context of multi-agent learning. For example, the authors may try to create some toy examples/experiments to demonstrate why the bayesian formulation is necessary and why the previous methods failed. Therefore, I'm not sure if the paper provides new insights/substantial technical contributions. - Following the same line, I think the paper should be better motivated in the next version. Before delving into the derivations (how to do bayesian MAIL), I don't quite understand why we should do bayesian MAIL. - It seems the theoretical analysis mainly discuss what difference will the approximation make when we do Nash Q learning. Although this is useful, this seems irrelevant to the main topic, i.e. multi-agent imitation learning (hence a gap between theory and practice). So the paper should also discuss more on what difference will it make in multi-agent imitation learning. Otherwise, the main claim should be the mean-field approximation. - I think a detailed experiment setup (environments illustration, tasks description, evaluation metrics, etc) is missing at the beginning of the experiments section. So I feel this part is a bit hard to follow. [1] A bayesian approach to generative adversarial imitation learning. [2] Multi-agent generative adversarial imitation learning.

Correctness: Yes.

Clarity: Yes. Most parts are easy to follow and the paper has a clear structure. See the weakness section for more comments.

Relation to Prior Work: Yes.

Reproducibility: Yes

Additional Feedback: Post-rebuttal feedback: I have read the author response and I think the authors have partially addressed my concerns, with a bit handwavy arguments. I hope the authors will address them in the next version in a more clear way. Overall, I choose to keep my original rating.


Review 3

Summary and Contributions: This paper introduces an approach for multiagent imitation learning (MAIL). The authors propose a new algorithm titled Bayesian Multi-type Mean Field Multi-agent Imitation Learning (BM3EIL), combining imitation learning with the multi-type mean field approximation approach of Subramanian et al., (2020). The authors introduce theoretical results establishing convergence guarantees of their approach in certain classes of games. Empirical evaluations are conducted in two domains (the first comparing Bayesian and non-Bayesian counterparts of the existing MA-DAAC algorithm, and the second comparing the proposed BM3EIL approach against several baselines). Please find my detailed comments below. For convenience, I have numbered the main points that I would appreciate the authors’ feedback on.

Strengths: In settings where access to an expert policy is available, imitation learning bears potential to have a significant impact on policy learning in multiagent settings. The approach, to my knowledge, is the first combining mean field theory, imitation learning, and attentional mechanisms into a single cohesive algorithm. The empirical results, albeit limited in scope, are promising in the sense that the proposed algorithm both improves final performance and reduces sample complexity of training.

Weaknesses: 1. The evaluation of the proposed approach (BM3IL) against existing baselines is only conducted on a single domain (the “transportation” environment). While the proposed approach exhibits higher learning rate and improved final performance compared to the baselines, the lack of a more thorough comparison makes it very difficult to judge the significance of the work. I am open to increasing my review score if the authors are able to add comparative results across additional new domains (namely, those which are characteristically different from the transportation environment already considered). 2. Moreover, a closely-related work that is not cited nor compared against is “Coordinated multi-agent imitation learning” (Le et al., 2017), and the closely-related “Data-Driven Ghosting using Deep Imitation Learning” (Le et al., 2017). I recommend the authors compare against this approach to better justify the significance of their work. 3. The interdependence of the agent ‘types’ and the resulting policy-level interactions is quite opaque here, and would benefit from significantly broader analysis. A few related remarks: a) Presumably, increasing the number of types has a negative impact on learning rate (this seems somewhat evident in Figure 4). Does it have a statistically significant impact on the final performance? How should practitioners choose the appropriate number of types? b) Do agents tend to specialize in one type? I.e., in equation 11, is $\alpha_m$ generally low or high entropy? c) How does this type-attention evolve throughout training (and likewise, throughout the state-action space for a fixed policy?)

Correctness: The methodology seems generally sound. The empirical claims are difficult to judge, given the B3MEIL approach is only evaluated on a single domain (see ‘weaknesses’ section).

Clarity: The paper is generally well-written, though suffers from a lack of clarity in some important sections: 4. [Equation 1] ] I believe the inner log in the right hand term of Equation (1) should not be present. I assumed it was a typo, but it is present throughout the text, even for the authors’ proposed approach (e.g., in Equation 3). If intentional, why is this necessary? 5. [Section 3.1.1] The paper introduces the problem scenario as a Markov game in Section 2.1; however, it introduces the notion of binary observations (which are a function of rewards here) in Section 3.1.1 (necessary for training a discriminator). This seems to suggest that perhaps the problem formulation should be corrected to a Partially Observable Markov game (POSG). 6. [Figure 4] While the distinction between the first and second subfigures were clear for B3MEIL, it was less so for the baseline algorithms. Are ‘types’ enforced for the other algorithms via simply partitioning the agents (presumably without any hierarchical attentional mechanism overlying those baseline policies?). If so, any conjectures as to why performance for the baselines drops when the number of types increases? Minor/typos: [l66] “Nash equilibrium [5] [describes] the situation that...” [l73] With respect to Nash-Q learning, the authors state that “The Q function will eventually converge to a Nash equilibrium”. However, this only holds for a restricted class of games, which should be clarified as such (e.g., see “Nash Q-Learning for General-Sum Stochastic Games”, Hu & Wellman, 2003 for details). [Section 2.2] Function D is undefined. (This is the GAIL discriminator, but should be clearly defined for unfamiliar readers). [l93] “Let the policy [be] parameterized by…” [Equation 11] Please add \qquads between the 3 expressions in this equation, as they are difficult to parse. [l199] “The mean field within each type can be computed through averaging the actions of each [agent]“ [l267] “”The experimental results showed that BM3IL outperforms 268 all other algorithms with faster convergence speed”

Relation to Prior Work: The distinction of the proposed work from prior contributions is fairly clear. Please refer to ‘weaknesses’ section regarding missing prior work/necessary comparisons.

Reproducibility: Yes

Additional Feedback: Post-rebuttal feedback: Overall, the authors have done a solid job of addressing my major concerns. Specifically, they have run new experiments on a new domain, conducted a more thorough analysis of the attention mechanism in their previous experiments, and fixed some noted mistakes in their equations. The effort the authors have already put into addressing my concerns is strong enough to convince me to increase my score to an accept.


Review 4

Summary and Contributions: The paper proposes a bayesian formulation for multi-agent imitation learning problem as well as the variant of the mean field approximation to make it scalable.

Strengths: * Strong experiments demonstrating a consistent improvement in sample efficiency for the proposed method against other baselines. * Theoretical results. * Good clarity of the paper

Weaknesses: Whereas Berlin seems like a complex environment, other methods do not seem to struggle much to learn it in fewer epochs than other environment. It suggests that the environment may not be that complex. It would be beneficial for the scalability claims to study the method in scenarios, where even the baseline struggles to learn.

Correctness: Yes.

Clarity: Well written

Relation to Prior Work: Yes.

Reproducibility: Yes

Additional Feedback: **Post-rebuttal** After having read the authors response as well as the other reviews, I am inclined to keep my current score.

[Author Response · NeurIPS 2020]

**Reviewer 1**   Q: Novelty against [3]: The differences: (1) They do reinforcement learning, while we do imitation learning, which is much harder. (2) We established a Bayesian formulation for MAIL to achieve sample efficiency. (3) The approximation form is different. We do both within-type and cross-type mean field, while they only do the former. Q: typos, and how the error bars were calculated Figures 3 & 4: Thanks for the suggestion! Will do. As shown in appendix, we run each algorithm multiple times and plot the mean and standard deviation (shaded area) of the results. Q: the clarity of the paper. For example, why to assume it is only necessary to model interactions within one type of agent (lines 151-152): Thank you for the suggestion. Will do. A quick answer is that agents of the same type have similar effects on another agent, and agents of different types show different effects (Eqs 6 and 7, lines 158-161). Q: MA-DAAC not scalable: The computation becomes expensive as number of agents increases. In our implementation, the computational complexity depends on the number of types, which is much smaller than the number of agents.

**Reviewer 2**   Q: why such a bayesian framework provides special advantages in the context of multi-agent learning..: Sample efficiency is a key issue in multi-agent imitation learning, because collecting samples in the real world is expensive. Bayesian parameter estimator can take full account of the uncertainties related the cost-function parameters $\phi$ compared to a point estimator, and is shown in Section 5.1 to improve sample efficiency by enhancing exploration. Q: The connection between the theory of Nash equilibrium and the method of the paper: Our paper is motivated by applying MAIL in real-world, where we focus on the challenge of sample efficiency and scalability. Multi-agent mean field is one method we introduce towards solving the scalability challenge. What we can show now is that with our approximation we can converge to Nash equilibrium, and other theoretical development will be future work. Q: missing a detailed experiment setup (environments illustration, tasks description, evaluation metrics, etc): Thanks for pointing out. We will give more details in the revision. Details for the Rover Tower can be found in line 249, citation [6]. Details for transportation environment can be found in [17]. Appendix 8.4 gives more details for both experiments.

**Reviewer 3**   Q: I am open to increasing my review score if the authors are able to add comparative results across additional new domains (not transportation): Thanks for your suggestions. The right figure shows the performance comparison in Cooperative communication environment (Multi-Agent Actor-Critic for Mixed Cooperative-Competitive Environments (Ryan et al., 2017)). We are also investigating other more complex environments — we haven't reached conclusions yet due to the extremely short time for rebuttal but will incorporate them in the final version.

Q: not compared against "Coordinated multi-agent imitation learning" (Le et al., 2017), "Data-Driven Ghosting using Deep Imitation Learning" (Le et al., 2017): Many thanks for pointing out the two papers! We will include them in the discussion. Actually one paper, "Coordinated multi-agent imitation learning", has been discussed in one of our benchmarking algorithm, "Multi-Agent Generative Adversarial Imitation Learning" (MA-GAIL), Section 6 citation 40.

Q: More insights into the connection among types, attention, and interactions: As shown in the right figure, where we plot the attention weight that agent type 0 put on other types in transportation environment with 10 types of vehicles. It shows that the agent learns to put different attention at different time step. In the morning (8 am) it puts more attention on 1-2 types of vehicles. In the evening (6 pm) it spreads it attention to more types of vehicles since more types of vehicles are moving around in the road network at this time.

Q: [Equation 1] typo: Thanks, it is a typo.

Q: [Section 3.1.1] Whether the formulation is a SG or POSG since it introduces the notion of binary observations (which are a function of rewards here): We are solving a Markov game, where each agent has full observation to the system state. We introduce the binary observations to delegate the reward, for the purpose of drawing the connection between imitation learning and probabilistic graphical models, and formulating as a Bayesian model. How we delegate the reward has nothing to do with the problem formulation. Q: why performance for the baselines drops when the number of types increases: One type refers to a subset of agents who have the same state and action spaces, and the same goal. Type is a virtual concept, which we introduced in our algorithm to improve scalability. Some of the baseline algorithms, such as MA-GAIL and MA-DAAC do not have this component. They work on each individual agents as suggested in their original paper. In Figure 4, as the number of types increase, there are more diverse behaviors of the agents (since different types have different goals), potentially more interactions, and hence more difficult to learn for the baseline algorithms.

**Reviewer 4**   Q: study more complex scenario where even the baseline struggles to learn: Thanks for your suggestions. We are investigating in more details and more complex environments, due to the limited short period of time in rebuttal, we are not able to get the results, but will make sure to incorporate more in the final version.

[Meta-Review · NeurIPS 2020]

All reviewers agree this paper is a clear accept. The most critical reviewer was satisfied by the authors' rebuttal addressing his major concerns as the authors have run new experiments on a new domain, conducted a more thorough analysis of the attention mechanism in their previous experiments, and fixed some noted mistakes in their equations.